# One Hundred and Fifty Years of Skin Effect

**Oldřich Coufal** 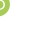

Department of Electrical Power Engineering, Brno University of Technology, Technicka 12,
61600 Brno, Czech Republic; coufal@vut.cz

**Abstract:** In 1873, J.C. Maxwell derived relations for current density and current in a solitary long cylindrical conductor supplied with variable current. According to Maxwell, the current density in a conductor increases towards the conductor surface. This phenomenon is called the skin effect. The skin effect affects in particular the inductance of a line formed by one or several long parallel conductors. A number of papers have been published regarding the skin effect and its effect on inductance. Another phenomenon, closely related to the skin effect, is the proximity effect, which refers to the dependence of the current density in a conductor on the proximity of other conductors through which a time-variable current is flowing. Many published papers deal with the calculation of current density in two conductors, using the method for calculating current density in a solitary conductor. All the phenomena given above can be analysed and quantitatively described based on the knowledge of current density in conductors, and therefore the method for calculating current density in a group of conductors is of fundamental significance. It follows from the analysis performed that the skin effect is not a general characteristic of current density in long conductors, except for in the solitary conductor. This conclusion affects the knowledge of the phenomena associated with the skin effect.

**Keywords:** transmission line; skin effect; proximity effect; self-inductance; induced voltage; equivalent circuit

## 1. Introduction

Current density in a conductor through which variable current passes increases towards the conductor surface. This phenomenon is called the skin effect. The skin effect is inextricably linked with current density. It is therefore strange that the existence of this phenomenon is generally assumed, even though an exact method for the calculation of current density was until recently known only for the infinitely long solitary conductor of circular cross section with constant resistivity. The aim of the present paper is an analysis of current density in long parallel conductors of arbitrary cross section with respect to the skin effect and other related phenomena. The starting point of the analysis has been the recently proposed accurate method for the calculation of current density in the conductors under investigation.

In 1873, J.C. Maxwell published the book "A Treatise on Electricity and Magnetism" [1]. The book is in two volumes and divided into articles; it was published in three editions [1–3]. The next editions, for example [4], are unabridged republications of the third edition [3]. Articles 689 and 690 in [1–3] deal with the calculation of current density and current in a solitary straight infinitely long conductor of circular cross section supplied with variable current. According to Maxwell, the amplitude of the current density of alternating current (of non-zero frequency) passing through a conductor increases towards the surface of the conductor. This phenomenon is called the skin effect.

The term skin effect is not used in [1–3]; it was probably introduced later. For example, according to [5], Horace Lamb published a paper in 1883 [6] which dealt with the motions of electricity produced in a spherical conductor by any electric or magnetic operations outside it, which was later to be known as the skin effect [5]. J.J. Thomson, who prepared the

third edition of Maxwell's book, added in the form of a footnote a derivation of the formula for calculating the inductance of a cylindrical conductor supplied with sinusoidal current. As part of this derivation, he also obtained a relation for the dependence of conductor resistance on the frequency of the supply current.

Another phenomenon closely related to the skin effect is the proximity effect, which refers to the dependence of current density in the conductor on the proximity of further conductors through which a time-varying current flows. Many published papers deal with the approximate calculation of current density in two conductors using the method for calculating current density in a solitary conductor. What is addressed in papers on the calculation of current density in two conductors is usually the skin effect, the proximity effect and the dependence of resistance on frequency, and their effect on self-inductance. Since the publication of the book [1], a plethora of articles and parts of books have been published that deal with the subject matter of the present review. In [7], published in 1959, the list of references contains 454 items, and one recent item is [8]. Thus, it is not possible to evaluate all the relevant papers.

All the phenomena given above can be analysed and quantitatively described based on the knowledge of current density in conductors, and therefore the method for calculating current density in a group of conductors is of fundamental significance. An accurate method for calculating current density in conductors forming a group has only recently been published [9–12]. The subject of investigation in [9–12] is a physical and mathematical model of a task referred to as the basic task.

The basic task is the calculation of the current density $J_{\text{bas}}(x, y, z)$, in a group of conductors that do not move and have the following properties;

- Each of them has a long part parallel to the axis $z$ in the coordinate system $xyz$;
- their cross section is arbitrary but does not change along the long part;
- some are connected at the end to a voltage or current source;
- some are connected at the end to other conductors in the group;
- the permeability of the conductors and their surroundings equals the vacuum permeability, $\mu_0$;
- they are placed in a magnetic field produced by a number of conductors that belong to the group or by an external source.

Solving the basic task is trivial if the magnetic field in which the conductors are placed does not change with time. In the case of a magnetic field varying with time, the solving of the basic task is problematic if some parts of the conductors are not parallel to $z$. In the physical model of the basic task it is assumed that

- the conductors in the group have only a long part, which is infinitely long, and each conductor is therefore determined by its cross section;
- the conductor resistivity, $\varrho$, is given and only depends on $x$ and $y$;
- the phenomena are quasi-stationary, i.e., the propagation speed of current and magnetic field is infinitely high;
- the displacement current and leakage current between conductors are neglected.

The physical model is based, in the first place, on Faraday's law of electromagnetic induction and is an application of the Biot and Savart law, loop current method, Kirchhoff's voltage law, and Ohm's law. Applying these laws to a finite segment of conductors yields a system of equations whose solution is a current density whose nonzero component is only the *z*-component:

$$J(x, y) = \langle 0, 0, J(x, y) \rangle.$$

In the mathematical model, the conductor cross sections are approximated by the $a(N, d)$ aggregate of disjunct rectangles $A_i$,

$$a(N, d) = \bigcup_{i=1}^{N} A_i,$$

in which only the resistivity, $\varrho_i$, and the current density, $J_i$, have constant values, and

$$d = \max_{i=1,2,\ldots,N} d_i,$$

where $d_i$ is the length of the diagonal of rectangle $A_i$. The mathematical model is a system of $N$ ordinary differential equations of the first order [9–12]. A detailed description of the physical and mathematical models is given in [9–12], together with the solution to specific cases. An exact solution to the mathematical model is the set

$$J_{\text{mat}} = \{J_1, J_2, \ldots, J_N\}. \tag{1}$$

$J_{\text{mat}}$ can also be considered a piecewise constant function, $J_{\text{mat}} = J_{\text{mat}}(x, y, d)$, defined in the plane $xy$:

$$J_{\text{mat}}(x, y, d) = \begin{cases} 0 & \text{for } (x, y) \notin a(N, d) \\ J_i & \text{for } (x, y) \in A_i. \end{cases} \tag{2}$$

It holds that

$$\lim_{d \to 0} J_{\text{mat}}(x, y, d) = J(x, y).$$

This means that $J_{\text{mat}}$ can approximate the solution $J(x, y)$ of the physical model, theoretically with an arbitrary accuracy. In practice, the accuracy is limited by the level of the calculation means used. For a sufficiently small $d$, $J_{\text{mat}}(x, y, d)$ is an accurate solution to the physical model and can be considered a solution to the basic task. The method for the calculation of the current density is numerical. In the calculation, the number of rectangles that approximate the cross sections of conductors is approximately 10,000. The accuracy of the calculation is assessed via comparing the results for different values of $d$.

Part of the calculation of current density in conductors is the calculation of independent magnetic fluxes. The determination of independent fluxes is described in [9–12]. If in the calculation of current density in a solitary conductor or in a loop formed by two conductors the conductor cross sections are symmetrical and the values of conductor resistivity are also symmetrical, the determination of independent fluxes is simple if symmetry is used [13–21].

## 2. Solitary Conductor Concept

In ref. [1], Art. 689 and 690, Maxwell proposed a method for the calculation of current density, $J_{\text{sol}}$, in a solitary long cylindrical conductor. In the following analysis of Maxwell's result, the quantities are denoted by currently used symbols. In ref. [1], $J_{\text{sol}}$ is the solution to a differential equation; it depends on the distance, $r$, from the conductor axis and on $t$,

$$J_{\text{sol}}(r, t) = -\frac{\mu_0}{4\pi\rho} \sum_{k=0}^{\infty} \left[ \frac{\mu_0 r^2}{4\rho} \right]^k \frac{i^{(k+1)}(t)}{(k!)^2}, \tag{3}$$

where

$$i(t) = i^{(0)}(t), \quad i^{(k+1)}(t) = \frac{\mathrm{d}^{k+1}}{\mathrm{d}t^{k+1}} i(t). \tag{4}$$

$i(t)$ is the given function and, according to (19) in [22], it holds that

$$J_{\text{sol}}(0, t) = \frac{\mu_0}{4\pi\varrho} i^{(1)}(t). \tag{5}$$

If the function $i(t)$ is sinusoidal, then the function $J_{\text{sol}}$ is also sinusoidal,

$$J_{\text{sol}}(r, t) = \hat{J}_{\text{sol}}(r) \sin[\omega t + \varepsilon(r)], \quad \omega = 2\pi f,$$

and its phasor, $\underline{J}_{\text{sol}}$, is a solution to the equation

$$\frac{\mathrm{d}^2 \underline{J}_{\text{sol}}}{\mathrm{d}r^2} + \frac{1}{r}\frac{\mathrm{d}\underline{J}_{\text{sol}}}{\mathrm{d}r} - \mathrm{j}\kappa^2 \underline{J}_{\text{sol}} = 0, \quad \kappa^2 = \frac{\omega\mu_0}{\varrho}, \tag{6}$$

which has been derived, for example, in ref. [16].

Using Maxwell's equations in differential form, an equation can be derived for the solitary conductor with an arbitrary cross section, which is satisfied by current density over the cross section of a long conductor,

$$\nabla^2 J - \mu_0\, \gamma\, \frac{\partial J}{\partial t} = 0, \tag{7}$$

where $\gamma$ is the electric conductivity of the conductor. Expressing current density in (7) via the vector potential, $A$, will yield an equation that is similar to (7).

The solution to Equation (6) can be expressed using the Bessel or the Kelvin functions [23], Art. 16.4. Since the publication of [1], a number of methods have been published based on the solitary conductor concept; some of the recent ones are, for example, [24–28].

**Example 1.** *Cylindrical Al conductor ($\varrho = 2.650 \times 10^{-8}\ \Omega{\cdot}m$ at $20\,°C$ [29]), whose axis is the axis z and the cross section radius $r_1 = 6\,mm$. The function $i(t)$ has been chosen such that it holds that $J_{\text{sol}}(0,t) = \hat{i}\sin\omega t$, where $\hat{i} = 1\,A{\cdot}m^{-2}$.*

The solution to Equation (6) in Example 1 is given in Figure 1.

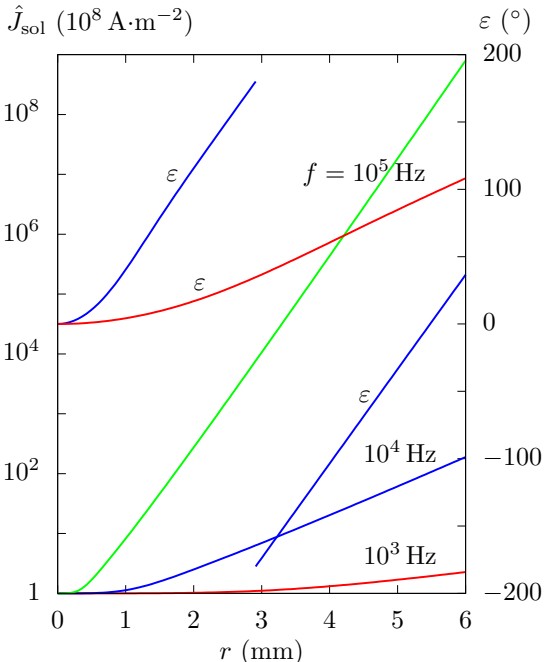

**Figure 1.** Solution to Equation (6) in Example 1. Dependence of amplitude, $\hat{J}_{\text{sol}}$, on the distance, $r$, from the conductor axis for $f = 10^3, 10^4$, and $10^5$ Hz; dependence of initial phase, $\varepsilon$, on $r$ for $f = 10^3$ Hz (red line) and for $f = 10^4$ Hz (blue lines).

The curve $\varepsilon(r)$ for $f = 10^4$ Hz in Figure 1 has two parts and is discontinuous. It is only a seeming discontinuity since $\varepsilon$ is modified such that $\varepsilon \in (-\pi, \pi]$, which is possible because the period of the function sin is $2\pi$; $\varepsilon(r)$ is continuous for $f = 10^3$ Hz. The number of seeming discontinuities of $\varepsilon(r)$ increases with increasing $f$. The function $\hat{J}_{\text{sol}}(r)$ is increasing (monotonous) in contrast to the function $J_{\text{sol}}(r)$, which is not monotonous, and for the discontinuous function $\varepsilon(r)$ it is oscillating. This property of current density is not limited

to the solitary conductor of circular cross section; it also occurs, for example, with the loop formed by two conductors of rectangular cross section. If the surface of the conductors is parallel with the planes $xy$ and $xz$, then the current density, $J(x, y)$, may oscillate for $y = \text{const}$, as can be seen in Figure 3 in ref. [21].

The amplitude, $\hat{J}_{\text{sol}}$, is the largest on the conductor surface, $r = r_1$, and it decreases with deceasing $r$; for $r = r_1 - \delta$, its value for high frequencies is e-times smaller than on the conductor surface, where $\delta$ is the so-called penetration depth [30]:

$$\delta = \sqrt{\frac{2\varrho}{\omega \mu_0}}. \tag{8}$$

Related to sinusoidal current density in a solitary conductor is the technical term skin effect, which should characterize current density in conductors. According to ref. [31], item 121-13-18, "Skin effect is a phenomenon in which the current density is greater near the surface than in the interior of the conductor for an alternating electric current in the conductor". In the literature are further definitions, which are either similar to the definition in ref. [31] or they define the skin effect as a phenomenon in which most of the current in the conductor flows through a thin layer near the conductor surface. In all the definitions, only the word 'conductor' is given, although all the definitions were proposed in analyses of the current density in a 'solitary cylindrical conductor'. Examples are given below that raise doubts as to the given skin effect definitions.

Figure 2 gives the current density $J_{\text{sol}} = J_{\text{sol}}(r, t)$ in Example 1 at five instants, $t = \theta T$, where $\theta = 0, 0.3, 0.4, 0.5$, and $T = 1/f, f = 10^4$ Hz.

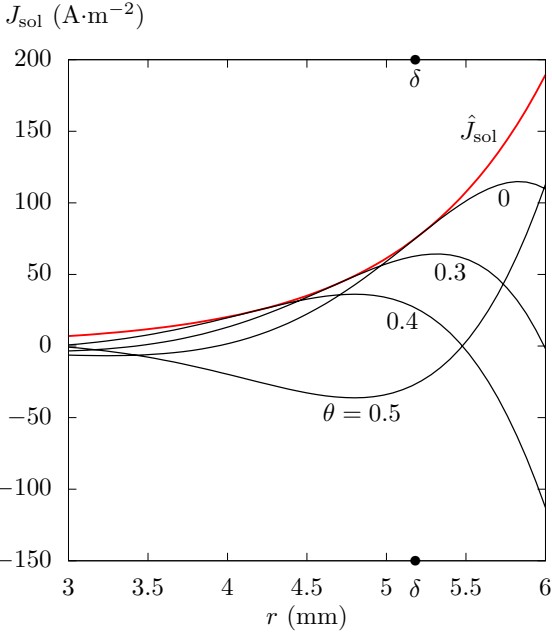

**Figure 2.** Dependence of instantaneous current density on $r$ for $t = \theta T$; $\theta = 0, 0.3, 0.4$, and $0.5$, in Example 1. The amplitude, $\hat{J}_{\text{sol}}$, is indicated by the red line and the quantity, $\delta$, is determined by Formula (8). For $r \in [0, 3)$ mm, the current density magnitude is smaller than $6.943\,\text{A·m}^{-2}$.

It follows from Figure 2 that for a solitary cylindrical conductor the conductor surface exhibits the maximum $\hat{J}_{\text{sol}}$ but not always the maximum current density. For a solitary conductor of rectangular shape, the situation is similar.

**Example 2.** *Solitary Al conductor ($\varrho = 2.650 \times 10^{-8}\,\Omega\text{·m at }20\,°C$ [29]) with the rectangle cross section $[0, 10]$ mm $\times$ $[0, 16]$ mm. The current through the conductor is*

$$I_{\text{rec}}(t) = \hat{I}_{\text{rec}} \sin \omega t, \text{ where } \hat{I}_{\text{rec}} = 1\,\text{A}, f = 10^4\,\text{Hz}.$$

This example is given, inter alia, because the current density,

$$J_{\text{rec}}(x, y, t) = \hat{J}_{\text{rec}}(x, y) \sin[\omega t + \varepsilon_{\text{rec}}(x, y)],$$

in a solitary rectangular conductor has not been calculated using the method proposed by Maxwell nor by methods derived from his method. The reason for this is ignorance of the initial and boundary conditions necessary for the solution of the respective differential equation. Figure 3 provides the dependence of the amplitude, $\hat{J}_{\text{rec}}$, on $x$ for several values of $y$ in Example 2. The values of $\hat{J}_{\text{rec}}$ were obtained by a method published in refs. [9–12], and in Figure 3 they are only given in one quarter of the conductor cross section because not only the amplitude but also the initial current density phase are symmetrical along the symmetry axes of the cross section $x = 5\,\text{mm}$ and $y = 8\,\text{mm}$.

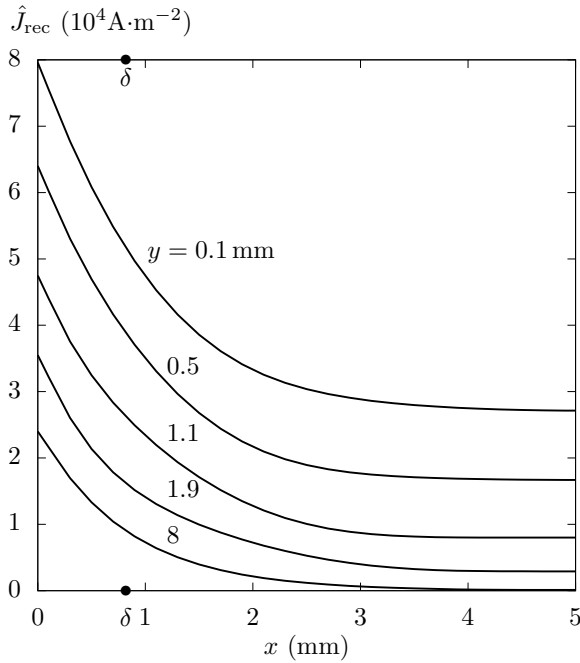

**Figure 3.** Dependence of the current density amplitude on $x$ and $y$ in Example 2 in a quarter of the conductor cross section; $\delta$ is defined by (8).

Figure 4 gives the current density in a quarter of the rectangular cross section of a solitary conductor at the instant $t = 0.39\,T$ in Example 2.

For a solitary tubular conductor with internal non-zero radius and constant resistivity, the skin effect obviously does not occur. $\hat{J}_{\text{sol}}$ is maximal and minimal near the outer and the inner surface, respectively; see Figures 2, 4, and 6 in ref. [16] and Figure 3 in ref. [17]. The skin effect is associated in practice not only with solitary conductors but also with two conductors, although in its definition (item 121-13-18 in ref. [31]) there is only 'conductor'. It does not occur in the loop formed by two coaxial tubular conductors; see Figures 3, 5, and 7 in ref. [13] and Figures 4, 5, and 6 in ref. [17]. In some cases, it does not occur even in the loop formed by two parallel conductors; see Figures 10 and 11 in ref. [18].

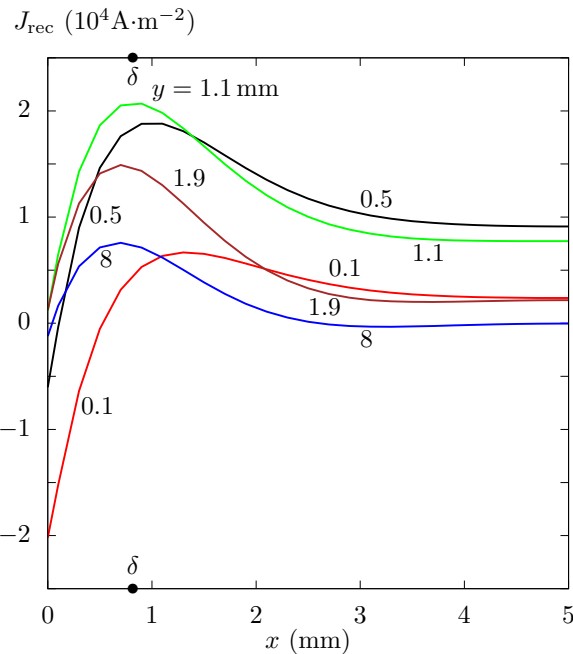

**Figure 4.** Instantaneous current density, $J_{\text{rec}}(x, y, t)$, for $t = 0.39\,T$ in Example 2.

## 3. Solitary Conductor versus Reality

Significant for practice are the theoretical models that are the most faithful image of reality. An infinitely long cylindrical conductor through which steady sinusoidal current flows is the basis of the solitary conductor concept. A conductor connected to the ideal current source can be considered solitary [32]. The ideal current source cannot be realized because, on a finite segment of the conductor, it produces an infinitely large voltage, but in spite of this a finite voltage is assumed in ref. [1], Art. 690. J.J. Thomson, who prepared the third edition [3] of [1], added in the form of a footnote a derivation of the formula for the calculation of inductance of a solitary conductor and the dependence of effective resistance on the frequency of the sinusoidal source the examined conductor is connected to. The skin effect is defined for the solitary conductor. Questioning the existence of solitary conductor also means questioning the skin effect.

Closest to the cylindrical conductor is in fact a pair of coaxial conductors connected to a source. The relation between current density in a solitary cylindrical conductor and in a pair of coaxial conductors is demonstrated in the following example.

**Example 3.** *A pair of coaxial conductors at a temperature of 25 °C, whose axis is the axis z, is formed by an inner cylindrical Cu conductor: cross section radius $r_1 = 10\,mm$, resistivity $\varrho_1 = 1.712 \times 10^{-8}\,\Omega{\cdot}m$ [29], and by an outer tubular Al conductor: inner radius $r_2 = 11\,mm$, resistivity $\varrho_1 = 2.709 \times 10^{-8}\,\Omega{\cdot}m$ [29], wall thickness $\Delta r$. A source of steady sinusoidal voltage with $f = 10^3\,Hz$ causes the drop $\hat{V} \sin \omega t$, where $\hat{V} = 1\,V{\cdot}m^{-1}$.*

The steady current density in coaxial conductors,

$$J_{\text{coa}}(r, t) = \hat{J}_{\text{coa}}(r) \sin[\omega t + \varepsilon_{\text{coa}}(r)],$$

is symmetrical with respect to the axis $z$; for the given parameters $r_1, r_2, \Delta r, \varrho_1, \varrho_2, f$, it only depends on the distance, $r$, from the axis $z$ and on $t$, and is directly proportional to the voltage drop on a segment of both conductors. According to refs. [13,18,22], it holds for a given voltage drop:

$$\lim_{r_2 \to \infty} \hat{J}_{\text{coa}}(r) = 0, \text{ for all } r. \tag{9}$$

It should be noted that (9) holds for the quasi-stationary state, which is assumed in the present review ( see Section 1), but assuming the quasi-stationary state is not adequate for a large $r_2$. According to ref. [33], the assumption of the quasi-stationary state is only adequate when the maximum dimension $\ell_{\max}$ of the system in the plane $xy$ is much smaller than the free space wavelength, i.e.,

$$\ell_{\max} \ll c/f, \text{ where } c \text{ is the speed of light.} \tag{10}$$

If the maximum magnitudes of $r_1$ and $\Delta r$ are several centimetres, then the inequality, (10) corresponds to the inequality $r_2 \ll c/f$.

In Figure 5, the function $\hat{J}_{\text{coa}}(r)$ in Example 3 is illustrated for three values of $\Delta r$. The method for calculating current density in two coaxial conductors was described in refs. [9,12–15]. In Example 3, $r_2 = 0.02c/f$ was chosen, i.e., $r_2 = 6000\,\text{m}$, in agreement with (10). It was established by calculation that for $\Delta r = 5\,\text{mm}$ and for all $r$, $\hat{J}_{\text{coa}}$ is smaller than $6.73 \times 10^5\,\text{A·m}^{-2}$ and $1.13 \times 10^3\,\text{A·m}^{-2}$ in the inner and outer conductor, respectively. The maximum value of $\hat{J}_{\text{coa}}$ for $r_2 = 6000\,\text{m}$ is only 3 % of the maximum value of $\hat{J}_{\text{coa}}$ for $r_2 = 11\,\text{mm}$.

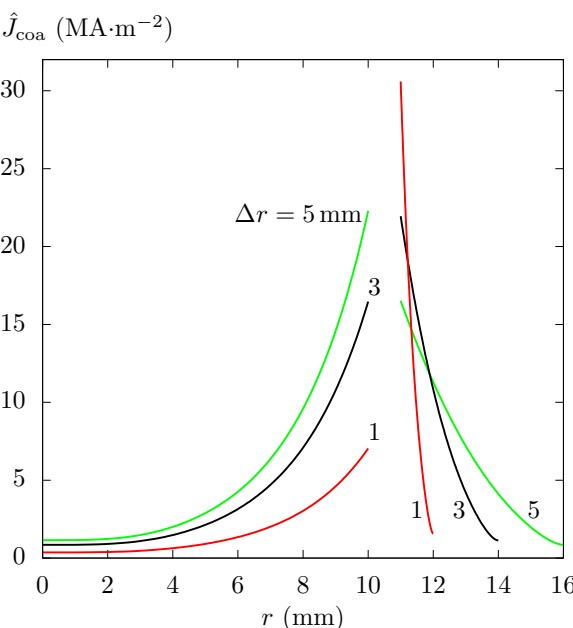

**Figure 5.** Dependence of amplitude, $\hat{J}_{\text{coa}}$, on $r$ in Example 3, for three values of $\Delta r$.

From the above, it follows that a solitary conductor connected to the ideal current source cannot be implemented. The inner conductor in a pair of coaxial conductors cannot be considered a solitary conductor either if the outer conductor is far from the inner conductor. The author of ref. [34] is of the opposite opinion when he says:"…the solitary conductor is a realizable abstraction. A solitary conductor is one where the radially varying inner electric and magnetic fields do not suffer any azimuthal variations, since no other neighbouring conductor exists. However, in a centred coaxial cable of long length, the shield conductor has no influence on the inner cylindrical conductor…". The author of ref. [34] does not say what effect on the inner conductor is concerned. If a magnetic field is concerned that is produced by coaxial conductors supplied by a dc source, then the author of ref. [34] is absolutely right. However, the subject of this review is the effect of the magnetic field produced by a source varying with time. Current density in conductors is a superposition of a source-driven current density and the density of eddy currents [12,14,15]. The eddy currents are determined by magnetic fluxes produced by the two conductors. The effect of the outer conductor on $\hat{J}_{\text{coa}}$ in the inner conductor in Example 3 can be seen in Figure 5. It

is difficult to imagine calculating current density in conductors that are not of circular cross section using the current density in a cylindrical conductor.

Current density in a conductor is connected with the vague term proximity effect. According to ref. [31], item 121-13-19, "The proximity effect is a non-uniform distribution of electric current density in a conductor, attributable to electric currents in neighbouring conductors". It is necessary to add that the cause of the non-uniform density of current in the conductor can also be the resistivity, which is not constant over the conductor cross section. If the cause of the non-uniform distribution of current density in the conductor is the current induced according to Faraday's law, then the induced current can also be produced by the conductor itself or by distant conductors, and also by, for example, a moving permanent magnet. The neighbouring conductor may be solitary, unconnected to the source, or it may form with the respective conductor a circuit in which there is a source, or it may form a part of a circuit with a source, or the neighbouring conductor is connected to the same pole of the source as the respective conductor. According to [9–12], the neighbouring conductor can be included in the calculation of current density in the respective conductor, and thus it is redundant to deal with the proximity effect.

### 4. Self-Inductance of a Pair of Conductors

Self-induction of a pair of conductors is characterized by the quantity self-inductance, $L$. $L$ determines the voltage induced in the conductors and is part of conductor impedance. There is no universal method for the calculation of $L$, because its value, among other things, depends on the current density in the conductors. The simple formulae given in the literature have been derived on the assumption of constant current density. This is a rather simplifying assumption because, in such a case, the induction phenomenon does not occur.

In the calculation of $L$, it is necessary to respect Formula (10), according to which the maximum distance of the conductors and the frequency of the current flowing through the conductors are limited. With a conductor distance of several metres, the frequency should be less than $10^7$ Hz. In ref. [13], the values are given for the energy in the magnetic and electric fields of coaxial conductors. From a comparison of these values, it results that neglecting the displacement current in the insulation between the conductors is acceptable for supply voltage frequencies not exceeding $10^7$ Hz if there is vacuum between the conductors. For a different insulation, the upper frequency limit would be lower. It is therefore necessary to assume the fulfilment of the condition $f < 10^7$ Hz; similar to ref. [35], p. 132, frequencies of up to 1 MHz are assumed so that quasi-stationary approximation may be assumed.

Formulae for the calculation of the self-inductance of a pair of long conductors are given in the literature, sometimes also with a derivation, e.g., [33,36–47]. Papers regarding the calculation of current density in a pair of long conductors often contain the method for calculating the self-inductance of a segment of the two conductors or the equivalent impedance of this segment [1,2,4,14,15,18,20,22,24,34].

Figure 6 gives the equivalent schematic of the segment of conductors between the planes $z = z_1$ and $z = z_2$, where $z_2 > z_1$.

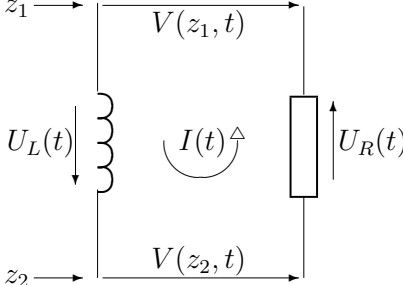

**Figure 6.** Equivalent schematic of a segment of two conductors between the planes $z = z_1$ and $z = z_2$. $U_L(t)$ is the induced voltage, $U_R(t)$ is the voltage across the segment of the two conductors. $V(z_1, t)$ and $V(z_2, t)$ are the voltages across the conductors for $z = z_1$ and $z = z_2$, respectively.

The inductor is characterized by the self-inductance coefficient $L$, which is used to calculate the voltage induced on the inductor,

$$U_L(t) = L\frac{\mathrm{d}I(t)}{\mathrm{d}t}, \tag{11}$$

where $I(t)$ is the current in the loop segment or in the circuit in Figure 6. The result of solving the basic task is the vector of current density in the loop conductors, and $I(t)$ is the flux of this vector through the conductor cross section. Solving the basic task thus also yields $I(t)$. If the voltage $U(t) = V(z_1, t) - V(z_2, t)$ is given, then Kirchhoff's voltage law holds for the loop and for the circuit in Figure 6,

$$U_L(t) = U(t) - U_R(t), \text{ where } U_R(t) = RI(t), \tag{12}$$

with $R$ being the given resistance of the loop segment. This means that determining $L$ is redundant, and in the equivalent diagram (Figure 6) the inductor characterized by $L$ is replaced by the voltage source $U_L(t)$. Unlike the calculation using $L$, the calculation of $U_L(t)$ using (12) is accurate, in particular in cases when the proximity effect manifests itself or when $U_L(t)$ is not sinusoidal.

In the literature, the calculation of $L$ is, in most cases, part of the calculation of the impedance of the circuit in Figure 6,

$$\underline{Z} = \Re(\underline{Z}) + \mathrm{j}\omega L, \tag{13}$$

where $\omega$ is the angular frequency of all the voltages and currents in the circuit. The application of (13) is conditional on the steady state of the circuit and on the distance of the conductors at which the proximity effect can be neglected. Another possible calculation of $L$ consists of replacing each of the conductors with a solitary conductor and replacing $L$ by the sum of internal and external inductance [43,48–52]. In the calculation of internal inductance, what is taken into account is only the magnetic flux inside the conductor and produced by this conductor. In the calculation of external inductance, only the magnetic flux between conductors is taken into consideration. This calculation procedure is mainly applied when the conductors are cylindrical because current density in solitary cylindrical conductors is easy to calculate.

The real component $\Re(\underline{Z})$ increases with increasing $\omega = 2\pi f$, and this is the reason for the incorrect statement "resistance increases with frequency", which is frequently given in the literature and applied in technical practice. Unfortunately, even in the definition of skin effect, in item 121-13-18 [31], there is a note 1: "The skin effect increases the resistance and decreases the inductance of a conductor with the frequency of the electric current". The standard [53] defines resistance (1) and resistance (2). Resistance (1) is determined by Ohm's law, which holds for metallic conductors over a large range of parameters. The dependence of resistance (1) on $f$ would above all mean the dependence of resistivity on $f$. According to ref. [40], Ohm's law holds for metallic conductors if $f \ll 10^{14}$ Hz. The value of resistance (2) is equal to $\Re(\underline{Z})$ according to [53], and therefore for $f > 0$ it is necessary to distinguish between $R$ and $\Re(\underline{Z})$.

## 5. Conclusions

The subject of the present paper is current density and current in a group of long conductors. The group is described in Introduction. The publication of Maxwell's book [1] marked the beginning of a one hundred and fifty year history of attempts at proposing a method for the calculation of current density in long parallel solid conductors.

Maxwell's method can only be applied to a solitary conductor connected to an ideal current source, which cannot be implemented. If it is assumed that the current density in the conductors of a loop formed by two conductors is the current density of two solitary conductors, then the accuracy of such an approximation increases with increasing distance of the conductors. In that case, the proximity effect cannot be

analysed, and the effect of current density on the result of the calculation of inductance is small because the decisive influence is that of the magnetic flux through the area between the conductors.

All the papers published up to now, with the exception of those by the author of this review and his fellow workers, deal with the solitary conductor, or they try to transfer the result of the solitary conductor to a group of conductors or they solve Equation (6) or (7). The current density calculated by these methods is approximate, and its application in the analysis of the proximity effect and in the calculation of inductance is therefore also approximate. The knowledge of exact current density obtained via solving the basic task makes it possible to quantify the proximity effect and thus not only to state that the current density in the conductor is affected by current in the neighbouring conductors. The proximity effect affects the voltage on the inductor, and the calculation of this voltage in terms of self-inductance is highly inaccurate and therefore it is suitable to characterize the inductor by a time-dependent induced voltage instead of self-inductance.

The definition in ref. [31] is special in that the skin effect is not defined for conductors but only for the conductor. On the other hand, it is very general because it does not contain any assumption as to the conductor length and shape. It is assumed in the definition that the current in the conductor is alternating, but in ref. [31], item 121-13-18, there is a note 2: "The skin effect occurs also in the more general case of any time-varying current". It follows from the specific cases quoted above that the skin effect is not an entirely general phenomenon, the more so because no analysis has so far been made of current density in conductors during transient phenomena and with conductor resistivity that is not constant over the conductor cross section. When using the term skin effect, it would therefore be advisable to state the relevant conditions: the form of the dependence of the voltage or current source on time, the shape of the conductor cross section, the position of the conductors in space, and in the case of transient phenomena, the initial current density. Playing with words in connection with the skin effect and the proximity effect can be replaced by an indisputable statement: "Current density is not constant in conductors that are in a time-varying magnetic field".

The skin effect is connected with the so-called resistance dependence on the frequency of the passing current. A strict distinction needs to be made between the resistance according to Ohm's law and the real component of impedance. For alternating current with non-zero frequency, the resistance according to Ohm's law does not depend on frequency and its value is not equal to the value of the real component of impedance, which depends on frequency.

Knowing the current density obtained by the solution of the basic task enables analysis of the phenomena connected with the skin effect, in particular analysing an accurate calculation of voltage on the conductor. This does not mean a refutation of the papers published so far. The methods for inductance calculation in these papers need to be taken as approximate but sufficiently accurate, in particular for larger conductor distances. Nevertheless, the accuracy of the results obtained by these methods can be obtained only via a comparison with the results obtained by the solution of the basic task.

**Funding:** This research work has been carried out in the Centre for Research and Utilization of Renewable Energy (CVVOZE). The author gratefully acknowledges financial support from the Ministry of Education, Youth, and Sports of the Czech Republic under Brno University of Technology specific research programme (project No. FEKT-S-23-8403).

**Institutional Review Board Statement:** Not applicable.

**Informed Consent Statement:** Not applicable.

**Data Availability Statement:** Data are contained within the article.

**Conflicts of Interest:** The author declares no conflict of interest.

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
