# Peer review of "One Hundred and Fifty Years of Skin Effect"

_applsci, doi:10.3390/app132212416_

Round 1

Reviewer 1 Report

Comments and Suggestions for Authors

1) The paper is very hard to read and follow, as the statements and focus jumps between different topics.
2) The main idea of the paper is not really clear. At first glance, it seems that the Author provides a review of different models for skin effect, then the focus drops on  tries to define the term "Skin effect" and how it differs in different papers. Then The author compares different the theoretical model to the practical one.
3) It is unclear, why Chapter 4 is included, as it doesn't define anything related to skin effect.
4) The conclusions are not conclusive, as no specific answer to what is incorrect in current skin effect calculation methods, how they evolved or how to improve it.

Currently, the paper seems to be a rough draft of various pieces of information about skin effect, some of with the author agrees with and others - not. I'd suggest clearing out the main purpose of the paper (whether it has to be a comparison of existing methods, evolution of methods or trying to demystify the terms in other papers) refining the way the information is presented.

Author Response

Dear Reviewer,

Thanks for you Reviewer report.

The subject of the investigation and the paper content are given in the Abstract
(lines 7-17) and in the Introduction (lines 38-44 in particular).
In other words: Skin effect is a term coined on the basis of the waveform of current
density in a solitary long conductor of circular cross section if the current flowing
through the conductor is not constant in time. The existence of a solitary conductor is problematic, yet the term skin effect is used as a characteristic of current density in arbitrary conductors that are connected to a source of time-dependent voltage or current. Another characteristic used is the proximity effect. The two characteristics are used without the exact distribution of current densities in the conductors being known. Dependent on current density is the value of self-inductance. The dependence of resistance on the frequency of the current flowing through the resistor is related to the skin effect. A method has recently been proposed for a correct calculation of the current density in long parallel conductors and thus the above phenomena and dependencies can be analysed for the given conductors. And all this is described in my paper. It is therefore no surprise that the manuscript has a Section 4. Also, it is not necessary to deal with methods for the calculation of skin effect, if the current density is known. Therefore, I would like to call attention to lines 326-335.
Unfortunately, your unfavourable review will probably be the decisive one.
I hope that the above brief overview will contribute to a more favourable opinion about my paper.

Sincerely,
Oldrich Coufal

Reviewer 2 Report

Comments and Suggestions for Authors

A throughly literature review about skin effect is given in this paper. The numerical solutions of current distribution are introduced generally. The modeling method of skin effect are discussed.

The overall quality of the paper is good. A minor comments is that on page 2, "in which only the resistivity roll and current density J", there is no equation using the roll and J nearby. Please check whether it is correct.

Comments on the Quality of English Language

N.A.

Author Response

Dear Reviewer,

Thanks for you Reviewer report.
As regards your comment:
Current density in conductors is the solution of a system of differential equations,
in which the function rho(x,y) also occurs. The corresponding system of  differential equations has been published in [9-12], as mentioned on line 53. An example of current density for a non-constant rho is given, for instance, in [20], Problem 3.

Sincerely,
Oldrich Coufal

Reviewer 3 Report

Comments and Suggestions for Authors

This is an interesting review mainly based from the author knowledge and papers he already publised in the past. It is well written and could be of interest for the audience of the journal. However, I have several comments that need to be taken into account before publication.

1.       While reading the paper, any reader could think that the method developed by the author is a purely analytical technique. When looking at the papers published by the author, it appears that the cross-sections of the conductors have to be discritized into small rectangles. So the technique uses some discretization step. The author should at least indicate this aspect and comment the impact of the discretization grid on the final results.

2.       Many comments and discussion in the paper are related to proximity effect but the paper does not really study this phenomenon and do not quantify the contribution of proximity effet in the different configurations studied in the paper. To my opinion, the discussion of proximity effect should be summarized in one single paragraph including the main contribution from the author.

3.       Some references related to the subject maybe included in the bibliography :

T. J. Higgins, "Formulas for the inductance of rectangular tubular conductors," in Electrical Engineering, vol. 60, no. 12, pp. 1046-1050, Dec. 1941, doi: 10.1109/EE.1941.6434575.

JabÅ‚oÅ„ski P, Szczegielniak T, Kusiak D, PiÄ…tek Z. Analytical–Numerical Solution for the Skin and Proximity Effects in Two Parallel Round Conductors. Energies. 2019; 12(18):3584. https://doi.org/10.3390/en12183584

Author Response

Dear Reviewer,

Thanks for you Reviewer report.

As regards comment 1.
The method for the calculation of the current density is given on lines 75-93.
After line 92, the following text has been added (in red):
The method for the calculation of the current density is numerical. In the calculation, the number of rectangles that approximate the cross sections of conductors is approximately 10000. The accuracy of the calculation is assessed via comparing the results for different values of d. 

As regards comment 2.
The results of investigating the skin effect are shown in Figs 1-5 and in comments
on these Figures. Furthermore, I refer to the Figures in my publications, as indicated on lines 165-172. The discussion of proximity effect is summarized in Sections 3 and 5.

As regards comment 3.
Both items have been included in the Bibliography (in red) and they are quoted on the respective places in the manuscript.

Sincerely,
Oldrich Coufal

Round 2

Reviewer 3 Report

Comments and Suggestions for Authors

My comments were taken into account. Thanks.

Comments on the Quality of English Language

Correct

Author Response

Thank you for your review.